# STABILIZED LIKELIHOOD-BASED IMITATION LEARNING VIA DENOISING CONTINUOUS NORMALIZING FLOW

## ABSTRACT

State-of-the-art imitation learning (IL) approaches, *e.g*, GAIL, apply adversarial training to minimize the discrepancy between expert and learner behaviors, which is prone to unstable training and mode collapse. In this work, we propose SLIL – Stabilized Likelihood-based Imitation Learning – a novel IL approach that directly maximizes the likelihood of observing the expert demonstrations. SLIL is a two-stage optimization framework, where in stage one the expert state distribution is estimated via a new method for denoising continuous normalizing flow, and in stage two the learner policy is trained to match both the expert's policy and state distribution. From the best of our knowledge, none of existing works solve the unstable training and mode collapse problem of GAIL. Experimental evaluation of SLIL compared with several baselines in ten different physics-based control tasks reveals superior results in terms of learner policy performance, training stability, and mode distribution preservation.

## 1 INTRODUCTION

Imitation learning (IL) (Abbeel & Ng, 2004; Ho & Ermon, 2016) aims to learn sequential decision-making policies directly from expert demonstrations, without access to reward signals from the environment. State-of-the-art (SOTA) IL approaches have primarily followed one of two paradigms: behavior cloning (BC) (Pomerleau, 1991) and generative adversarial imitation learning (GAIL) (Ho & Ermon, 2016). While both BC and GAIL have been studied extensively, each has crucial limitations. On the one hand, BC approaches employ supervised learning, which requires a large amount of expert demonstrations to avoid compounding errors due to covariate shifts (Ross et al., 2011; Ross & Bagnell, 2010). On the other hand, GAIL approaches (Jena & Sycara, 2020; Fei et al., 2020; Li et al., 2017; Hausman et al., 2017; Fu et al., 2017; Ke et al., 2019; Ghasemipour et al., 2019; Zhang et al., 2020; Arjovsky et al., 2017) connect IL with generative adversarial networks (GAN) (Goodfellow et al., 2014), but adversarial training processes are intrinsically unstable (Jena & Sycara, 2020; Arjovsky et al., 2017) and prone to mode collapse (especially when learning from multi-mode expert demonstrations) (Ghasemipour et al., 2019; Fei et al., 2020; Ke et al., 2019). To see this, consider Fig. 1, which shows an example of different IL algorithms on the Reacher task (Todorov et al., 2012; Hausman et al., 2017) with four targets (Fig. 1a). While the expert tends to visit all targets equally (Fig. 1b), the policy learned by GAIL is mode collapsed, primarily visiting the green one (Fig. 1e).

**SLIL: Overview and Contributions.** In this work, we are motivated to design an IL methodology with a stabilized training process and mode distribution preservation from expert demonstrations. We propose SLIL – Stabilized Likelihood-based Imitation Learning – a novel IL approach that directly maximizes the likelihood of observing the expert demonstrations. SLIL is a stable two-stage optimization framework, with the $1^{st}$ stage focusing on accurately estimating the expert state distribution, and the $2^{nd}$ stage training the learner policy to match the expert's policy and state distribution. Fig. 1c shows a quick view of our results: SLIL leads to a learner policy preserving the mode visitation distribution of the expert demonstrations. Our key contributions are as follows:

- We develop the first likelihood-based IL methodology that tackles the training instability issue of GAIL. SLIL's two-stage optimization framework is based on a tight lower bound on the joint policy and state distribution training objective from the likelihood-based IL formulation (Sec 4.1).

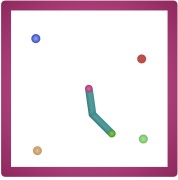
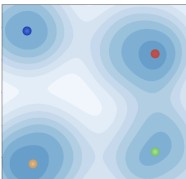
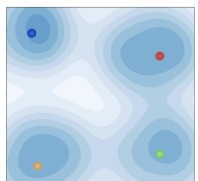
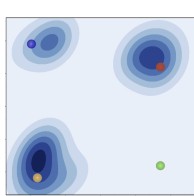

(a) Reacher4 with four mode targets.   (b) Mode coverage of expert policy.   (c) Mode coverage of SLIL policy.   (d) Mode collapse of DRIL policy.

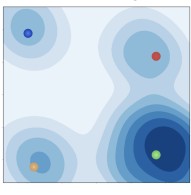
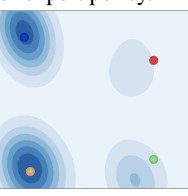
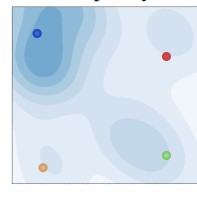

|  | EMD |
|---|---|
| SLIL | **0.33** |
| DRIL | 0.55 |
| GAIL | 0.41 |
| PWIL | 0.48 |
| Soft-SLIL | 0.67 |

(e) Mode collapse of GAIL policy.   (f) Mode collapse of PWIL policy.   (g) Mode collapse of Soft-SLIL policy.   (h) Learner policy EMD result.

Figure 1: Example results obtained by SLIL (Ours) and baselines on mode coverage. (a): A Reacher task, with four targets in different colors. (b)-(g) show the mode coverage (*i.e*, state distribution) with expert policy (b), our SLIL policy (c), DRIL policy (d), GAIL policy (e), PWIL policy (f), and SLIL implemented with SoftFlow (Soft-SLIL) (g). (h) show the earth mover's distance (EMD) (Ling & Okada, 2007) between expert and learner policy state distributions. All the distributions are visualized using kernel density estimation (KDE) (Sheather & Jones, 1991). None of the compared approaches solve the mode collapse problem.

- We propose the denoising continuous normalizing flow (DCNF) algorithm to accurately estimate the expert state distribution in the $1^{st}$ stage of SLIL to preserve expert modes. DCNF maps the expert state to the Gaussian noise distribution with a neural ODE, and trains the ODE using perturbed expert states (Sec 4.2).
- Our evaluation on ten different physics-based control tasks reveals that SLIL obtains superior results compared with SOTA baselines in terms of learner policy performances, training stability, and mode distribution preservation (Sec 5).[1]

## 2 RELATED WORK

**Behavior Cloning (BC)** (Pomerleau, 1991; Bohg et al., 2020) approaches for IL learn the expert policy via maximizing expert trajectory likelihood in the demonstration data, *i.e*, maximizing expert action likelihood. Though effective with abundant demonstrations, BC suffers from the covariate shift problem with limited data (Ross et al., 2011; Ross & Bagnell, 2010). Brantley et al. (2019) aims to address this via first pre-training an ensemble of BC policies, and then using reinforcement learning (RL) (Sutton & Barto, 2018) to train a learner policy whose cost is proportional to ensemble policies' prediction variance. In SLIL, we instead strive to solve the covariate shift problem via expert state-action distribution matching, learning the expert state distribution explicitly.

**Generative Adversarial Imitation Learning (GAIL)** (Ho & Ermon, 2016) employs the GAN framework (Goodfellow et al., 2014) to minimize the discrepancy between expert and learner state-action distributions. It jointly trains a generator (*i.e*, learner policy) to imitate the expert behaviors, and a discriminator (*i.e*, reward signal) to distinguish the state-action pair distributions between the expert and the learner. In GAIL, the discrepancy between the behavior distributions of the expert and the learner is measured by JS divergence. Using the variational lower bound of an $f$-divergence measure, several studies (Ke et al., 2019; Ghasemipour et al., 2019; Fu et al., 2017; Nowozin et al., 2016; Arumugam et al., 2019; Zhang et al., 2020) have extended GAIL from JS to a pre-defined $f$-divergence, *e.g*, KL (Fu et al., 2017), Reverse KL (Ke et al., 2019), Total Variation (Ross et al., 2011). However, all these approaches adversarially train the generator and the discriminator as a minimax game to reach an equilibrium, which can lead to training instability and mode collapse (Schroecker et al., 2019; Ghasemipour et al., 2019; Fei et al., 2020; Ke et al., 2019).

---

[1]For reproducibility, the code for our experiments is available at `https://www.dropbox.com/sh/buhqre6hfwdmvwu/AAAkc7SSjqhmtfalqqj5I7Iba?dl=0`

Several recent works try to avoid adversarial training in IL (Dadashi et al., 2020; Liu et al., 2020; Kim et al., 2020b; Rhinehart et al., 2018; Schroecker et al., 2019). To do so, Primal Wasserstein IL (PWIL) (Dadashi et al., 2020) considers the primal form of Wasserstein distance to match learner's and expert's state-action distributions. Neural Density Imitation (NDI) (Kim et al., 2020b) estimates expert's occupancy measure using which as a reward for maximum occupancy entropy reinforcement learning. Energy-Based IL (EBIL) (Liu et al., 2020) stems from Max-Entropy IRL (Ziebart et al., 2008) and estimates a surrogate reward function with score matching from expert demonstrations. Imitative Models (IM) (Rhinehart et al., 2018) learns a flow model that assigns high likelihoods to expert-like trajectories for test time goal-directed planning. GPRIL (Schroecker et al., 2019) applies masked autoregressive flows (Papamakarios et al., 2017) to learn predecessor state-action distribution in each training iteration, which adds complexity to learner policy training and requires training stabilization. Unlike these methods, we maximize expert state-action likelihood and strive to not only avoid adversarial training, but solve the mode collapse and unstable training problems as well.

## 3 LIL PROBLEM FORMULATION

**Notations.** We denote $\mathcal{S}$ as a set of states, $\mathcal{A}$ as a set of actions, $\mathcal{P} : \mathcal{S} \times \mathcal{A} \times \mathcal{S} \mapsto [0, 1]$ as the transition probability distribution, $r : \mathcal{S} \times \mathcal{A} \mapsto \mathbb{R}$ as the reward function, $\rho_0 : \mathcal{S} \mapsto \mathbb{R}$ as the distribution of the initial state $s_0$, and $\gamma \in [0, 1]$ as the discount factor. An agent makes decisions following a policy $\pi : \mathcal{S} \times \mathcal{A} \mapsto [0, 1]$, which specifies a probability distribution of choosing an action $a \in \mathcal{A}$ at a state $s \in \mathcal{S}$. With $s_0 \sim \rho_0$, then over time $t$, $a_t \sim \pi(a_t|s_t)$ and $s_{t+1} \sim \mathcal{P}(s_{t+1}|s_t, a_t)$ according to the policy $\pi$. We will denote the expert policy as $\pi_E$, and the learner policy as $\pi$. We denote $P_\pi(s, a)$ as the probability of observing a state-action pair $(s, a)$ when executing the learner policy $\pi$, and denote $P_E(s, a)$ to represent $P_{\pi_E}(s, a)$ for brevity. Moreover, for a function $h(s, a)$ of interest, we use an expectation over a policy $\pi$ to denote an expectation with respect to the trajectories it generates, *i.e,* $\mathbb{E}_\pi[h(s, a)] \triangleq \mathbb{E}[\sum_{t=0}^\infty \gamma^t h(s_t, a_t)]$.

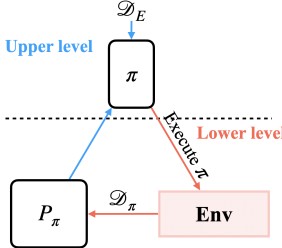

Figure 2: LIL framework.

**Likelihood-based Imitation Learning (LIL).** To avoid the instability from adversarial training (*e.g,* in GAIL), we introduce a *likelihood-based imitation learning* paradigm to learn the policy $\pi$ by directly maximizing the likelihood of the state-action pairs from the expert demonstration data $\mathcal{D}_E$. The LIL objective can be formally modeled as a bilevel optimization problem (Sinha et al., 2017):

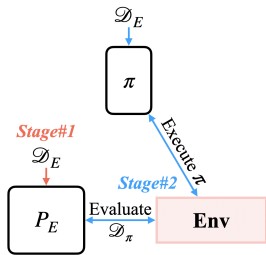

$$\max_\pi \mathbb{E}_{(s,a) \in \mathcal{D}_E}[\log P_\pi(s, a)], \text{ s.t. } P_\pi = \arg\max_P \mathbb{E}_{(s,a) \in \mathcal{D}_\pi}[\log P(s, a)], \quad (1)$$

Figure 3: SLIL framework.

with $P_\pi(s, a)$ as the probability of observing an expert state-action pair $(s, a) \in \mathcal{D}_E$ when executing the learner policy $\pi$. Notice that instead of maximizing trajectory likelihood as in BC, LIL maximizes the likelihood of observed state-action pairs. The variable function $P_\pi(s, a)$ can be learned from the lower-level optimization to maximize the log-likelihood of observing state-action pairs $(s, a) \in \mathcal{D}_\pi$ collected by interacting with the environment using the learner policy $\pi$ (*i.e,* the upper-level variable function). Moreover, since $P_\pi(s, a) = \pi(a|s) P_\pi(s)$, with $\pi(a|s)$ as the learner policy and $P_\pi(s)$ as the state distribution when executing $\pi$, the LIL objective can be rewritten as

$$\max_\pi \mathbb{E}_{(s,a) \in \mathcal{D}_E}[\log \pi(a|s)] + \mathbb{E}_{s \in \mathcal{D}_E}[\log P_\pi(s)], \quad \text{s.t.} \quad P_\pi = \arg\max_P \mathbb{E}_{s \in \mathcal{D}_\pi}[\log P(s)]. \quad (2)$$

**Challenges.** The LIL solution framework alternates between *policy learning* (*i.e,* the upper-level optimization to update $\pi$) and *state distribution estimation* (*i.e,* the lower-level optimization to learn $P_\pi$), as shown in Fig. 2. In each iteration, the learner policy $\pi$ is used to interact with the environment to collect $\mathcal{D}_\pi$. A normalizing flow is trained to estimate the state distribution $P_\pi$ of policy $\pi$ using $\mathcal{D}_\pi$ (*i.e,* lower-level optimization). Then, the policy $\pi$ is updated based on the state distribution $P_\pi$ using expert data $\mathcal{D}_E$ (*i.e,* upper-level optimization). However, two key challenges prevent a straightforward application of LIL. *C#1 Training stability:* It is challenging to properly grow $\pi$ and $P_\pi$ jointly to speed up and stabilize the training process in the bilevel optimization framework of Eq.(2). *C#2 Manifold hypothesis:* Normalizing flows are only valid when the data distribution and

the target noise distribution have the same dimensions, which conflicts with the observation that real-world data usually concentrate on low dimensional manifolds in a high dimensional space (*a.k.a.* the ambient space) (Papamakarios et al., 2017; Kim et al., 2020a; Song & Ermon, 2019; Ho et al., 2020; Belkin & Niyogi, 2003). Given an expert state distribution supported on a low dimensional manifold, the LIL framework will fail to estimate the ground-truth state distribution $P_\pi$ and fail to learn an expert-like learner policy $\pi$ efficiently.

To address these challenges, we propose the Stabilized Likelihood-based Imitation Learning (SLIL) framework shown in Fig. 3. SLIL relaxes the LIL problem to a two-stage optimization problem for improved training stability, and learns expert state distribution with denoising continuous normalizing flow (DCNF) to overcome the manifold hypothesis challenge. We formalize this methodology next.

# 4 OUR SLIL APPROACH

## 4.1 SLIL WITH EXPERT STATE DISTRIBUTION

As we discussed earlier, the training instability of the LIL framework in Eq.(2) comes from the joint training of policy $\pi$ (in the upper-level objective) and the corresponding state distribution $P_\pi$ (in the lower-level objective). In Theorem 4.1 below, we show that the LIL problem (as a bilevel optimization problem) can be relaxed to a two-stage optimization problem, which estimates the expert state distribution $P_E$ (in *Stage #1*), and trains the learner policy $\pi$ based on $P_E$ (in *Stage #2*).

**Theorem 4.1.** *The optimal objective of the likelihood-based imitation learning (LIL) problem in Eq.(2) is lower bounded tightly by the optimal objective of the following two-stage optimization problem (see our proof in Appx. A):*

$$\max_\pi \mathbb{E}_{(s,a)\in\mathcal{D}_E}[\log \pi(a|s)] + \mathbb{E}_{s\in\mathcal{D}_\pi}[\log P_E(s)], \quad s.t. \quad P_E = \arg\max_P \mathbb{E}_{s\in\mathcal{D}_E}[\log P(s)]. \quad (3)$$

Based on Theorem 4.1, we can relax the LIL problem by maximizing the two-stage optimization problem in Eq.(3).

**The solution framework of SLIL.** To solve the two-stage objective in Eq.(3), we model the learner policy $\pi$ as a deep neural network parameterized by $\theta$, and the expert state distribution $P_E$ using our DCNF parameterized by $\omega$. The training process involves two stages, described in Fig. 4: *(Stage #1) expert state distribution matching*, where we train an expert state distribution $P_{E\omega}$ based on DCNF to tackle the manifold hypothesis challenge (see details in Sec. 4.2); *(Stage #2) policy learning*, where we use the learned expert state distribution $P_{E\omega^*}$ to train the learner policy $\pi_\theta$ with the objective in Eq.(3). Note that updating the policy $\pi_\theta$ with $P_{E\omega^*}$ guidance involves action sampling. Thus, we follow an RL approach and employ proximal policy gradient (PPO) (Schulman et al., 2017) for back-propagation, with the Adam optimizer (Kingma & Ba, 2014) for updating the policy $\pi_\theta$. Comparing to the bilevel optimization framework of LIL in Fig. 2, the stabilized two-stage optimization framework for SLIL is shown in Fig. 3.

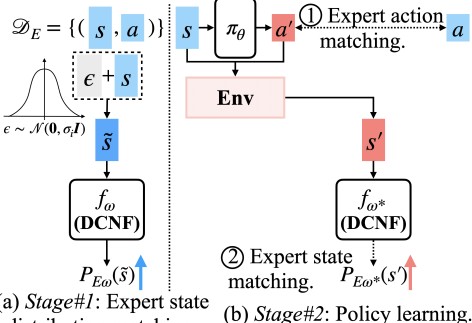

(a) *Stage#1*: Expert state distribution matching.  (b) *Stage#2*: Policy learning.

Figure 4: Illustration of our SLIL framework.

The objective function in Eq.(3) has two components: expert policy matching (*i.e*, BC), and expert state distribution matching (with $P_{E\omega^*}$). In each training iteration, the learning speed between the two components matter, and have impacts on the training stability. Similar observations are made in (Lee et al., 2018). To further improve the training stability, we introduce a gating function $\mathbf{1}_\lambda(\cdot)$ with parameter $\lambda$, which governs whether to run the second component of matching expert distribution in a particular iteration, *i.e*, $\mathbf{1}_\lambda(i) = 1$ iff $(i \mod \lambda) = 0$ for any training iteration $i$. For example, $\lambda = 2$ means that the second component is included in all training iterations with even sequence numbers. A lower $\lambda$ leads to more iterations to match the expert distribution, which can mitigate the compounding errors from BC. On the other hand, a higher $\lambda$ encourages supervision from expert policy matching (*i.e*, through BC). Therefore, with the gating function, the objective in Eq.(3) can be rewritten as

$$\mathbb{E}_{(s,a)\in\mathcal{D}_E}[\log \pi_{\theta_i}(a|s)] + \mathbf{1}_\lambda(i)\mathbb{E}_{s\in\mathcal{D}_{\pi_{\theta_i}}}[\log P_{E\omega^*}(s)], \quad \forall i. \quad (4)$$

## 4.2 DCNF FOR LEARNING EXPERT STATE DISTRIBUTION

There have been many solutions proposed to estimate the data distribution from a real world dataset $\{x\} \in \mathcal{X}$. The discrete normalizing flow (DNF) and continuous normalizing flow (CNF) methods both have advantages as explicit models for expressing data sample likelihood via the change of variable formula (Rezende & Mohamed, 2015). Discrete versions of normalizing flows (NF) (Rezende & Mohamed, 2015; Dinh et al., 2016; Kim et al., 2020a; Dinh et al., 2014) feature a sequence of $n$ bijective mappings, *i.e*, $f_\omega = f_n \circ f_{n-1} \circ \cdots \circ f_1$ and require computing the Jacobian of $f_\omega$ for back-propagation. These features restrict DNF's modeling capability as $f_\omega$ needs to be a once-differentiable bijection, and makes it computationally costly as the Jacobian matrix requires large computation power (Chen et al., 2018; Grathwohl et al., 2018). In contrast, CNF (Chen et al., 2018; Grathwohl et al., 2018) views the transformation between Gaussian noise and data samples as an ordinary differential equation (ODE) determined by $f_\omega$, which does not have model restrictions (Grathwohl et al., 2018), and expresses data sample density via the instantaneous change of variable formula (Chen et al., 2018). Therefore, CNF, *e.g*, FFJORD (Grathwohl et al., 2018), is more promising than DNF in terms of estimating the expert state distribution $P_{E\omega}$, where the ODE system function $f_\omega$ learns a bijection between random noise $z \in \mathcal{Z}$ and expert state $s \sim P_E(s)$, *i.e*, $f_\omega : \mathcal{Z} \mapsto \mathcal{S}$. The ODE and the expert distribution density follow the instantaneous change of variable formula, *i.e*,

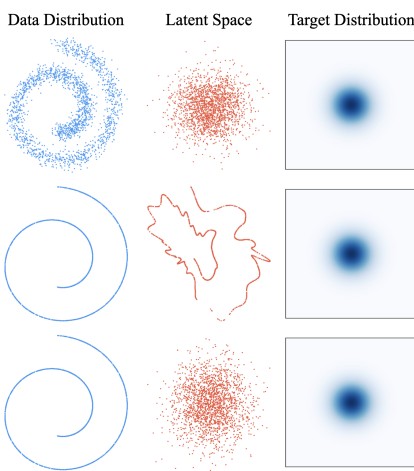

Data Distribution   Latent Space   Target Distribution

Figure 5: Illustration of FFJORD trained on 2D manifold (top), on 1D manifold (middle), and our DCNF trained on 1D manifold (bottom).

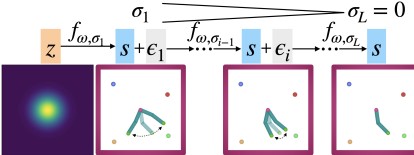

Figure 6: Illustration of DCNF.

$$\frac{z(t)}{t} = f_\omega(z(t), t), \text{ with } z(t_0) = z, \ z(t_1) = s \in \mathcal{S}; \log P_{E\omega}(s) = \log p(z(t_0)) - \int_{t_0}^{t_1} \text{Tr}\left(\frac{\partial f_\omega}{\partial z(t)}\right).$$

**Manifold hypothesis challenge.** The manifold hypothesis, *i.e*, real-world data tend to concentrate on a low dimensional manifold in a high dimensional space (Roweis & Saul, 2000; Belkin & Niyogi, 2003), has been observed on many datasets. As a result, since CNF is only valid when the data distribution and the target noise distribution have the same dimensions, the CNF will fail to estimate the ground-truth state distribution $P_E(s)$ if the expert states lie in a low dimensional manifold.

Fig. 5 shows examples to illustrate the manifold hypothesis challenge. The goal is to transform the data distribution displayed on the left column to the target Gaussian distribution on the right column using CNF. The transformed latent variable scatter plots (from the data distribution) based on FFJORD (Grathwohl et al., 2018) and our DCNF (described below) are placed in the middle column. It is clear that when the true data populates a 2D manifold over a 2D space, its transformed latent variables match a target Gaussian distribution well (the 1st row). On the other hand, when the true data resides in a 1D manifold of the 2D space, its transformed latent variable does not match the target Gaussian distribution (the 2nd row). This example illustrates how the manifold hypothesis deteriorates CNF's performance: when the data distribution lies in a low-dimension manifold of the ambient space, no homeomorphism can be easily created (Dupont et al., 2019; Kim et al., 2020a).

**DCNF.** Our experimental results in Fig. 5 (the 3rd row) and SOTA works (Kim et al., 2020a; Liu et al., 2019; Song & Ermon, 2019; Ho et al., 2020) have each observed that perturbing data with random Gaussian noise $\mathcal{N}(0, \sigma^2 \boldsymbol{I})$ (with $\sigma \geq 0$ as the noise level) can tackle the manifold hypothesis challenge, since the perturbed data will expand the data (*i.e*, expert states) from a low-dimension manifold to a high dimensional ambient space. It can also increase model generalizability given limited amounts of expert data. Denote a perturbed expert state $s \sim \mathcal{D}_E$ as $\tilde{s}$, with $\tilde{s} = s + \sigma\epsilon$ where $\epsilon \sim \mathcal{N}(0, I)$. When the noise level is very small, *i.e*, $\sigma \to 0$, $\tilde{s}$ approaches $s$. Therefore, we decrease the noise level $\sigma$ in each iteration when training the ODE system function $f_w$ of the expert state distribution $P_E$. Fig. 6 and Alg. 1 give our denoising continuous normalizing flow

(DCNF) algorithm. We first pre-define an arithmetic sequence of decreasing noise levels $\{\sigma_i\}_{i=1}^{L}$ whose common difference of successive numbers is $\frac{\sigma_1}{L}$, where $L$ is the number of training iterations. For each training iteration, we sample $B$ states $s_j$ from expert demonstrations $\mathcal{D}_E$, and $B$ random noises $\epsilon_j \sim \mathcal{N}(0, \sigma_j^2 \boldsymbol{I})$ with $j = 1, \cdots, B$ (line 2&3). The sampled noises $\epsilon_j$ are applied to the sampled states $s_j$ to obtain perturbed states $\tilde{s}_j$ (line 4). The perturbed states $\tilde{s}_j$ are input into the expert state distribution $P_{E\omega}$ to be evaluated, whose value will be used to compute the gradient with Adam (Kingma & Ba, 2014) to update $\omega$ (line 5). After training, a learned expert state distribution $P_{E\omega^*}$ (governed by the learned $f_{\omega^*}$ function) is obtained and used for learner policy training.

---

**Algorithm 1** Denoising Continuous Normalizing Flow (DCNF)

---

**Require:** Initial parameters $\omega_0$ for the ODE system function $f_\omega$ of $P_{E\omega}$; expert demonstrations $\mathcal{D}_E$ containing state-action pairs; predefined initial noise level $\sigma_1$, total number of training iteration $L$ and batch size $B$.

**Ensure:** The ODE system function $f_{\omega^*}$ of the expert state distribution $P_{E\omega^*}$.

1: **for** each epoch $i = 1, 2, \cdots, L$ **do**
2:      Sample $B$ states $s_j \in \mathcal{D}_E$ where $j = 1, \cdots, B$.
3:      Sample $B$ random noises $\epsilon_j \sim \mathcal{N}(\boldsymbol{0}, \sigma_i^2 \boldsymbol{I})$ where $j = 1, \cdots, B$ and $\sigma_i = \sigma_1 - \frac{\sigma_1}{L} \cdot (i - 1)$.
4:      Apply noise on sampled states $\tilde{s}_j = s_j + \epsilon_j$.
5:      Update $\omega_i$ to $\omega_{i+1}$ by ascending with the gradients: $\Delta_{\omega_i} = \sum_{j=1}^{B} \nabla_{\omega_i} \log P_{E\omega_i}(\tilde{s}_j)$.
6: **end for**

---

## 5   EXPERIMENTAL EVALUATION

To evaluate our proposed SLIL methodology, we conduct experiments on ten physics-based control tasks, including CartPole (Barto et al., 1983), Reacher (with 1, 2 and 4 targets), Hopper, Walker, HalfCheetah (with 1 and 2 running directions), Ant, and Humanoid all simulated with MuJoCo (Todorov et al., 2012). From these experiments, we show that: i) learner policy from our SLIL avoids mode collapse, by accurately preserving the expert mode distribution; ii) our DCNF can properly address the manifold hypothesis challenge in estimating the expert state distribution; iii) learner policies from SLIL have comparable or better performance than IL baselines; and vi) SLIL has more stability to hyper-parameter changes than IL baselines using alternative training (e.g., GAIL).

**Implementation Settings and IL Baselines.** We obtain expert policies of all tasks by running TRPO (Schulman et al., 2015a) with their ground-truth reward functions defined in the OpenAI Gym (Brockman et al., 2016). Then, we use the expert policies to generate expert demonstrations. We use Reacher with two and four target modes, and HalfCheetah with two target modes (Todorov et al., 2012) respectively (*i.e*, *Reacher2*, *Reacher4* and *HalfCheetah2*) to analyze the mode coverage of our SLIL vs IL baselines. Each expert policy obtained by TRPO has a particular distribution of reaching different target modes. We relegate more implementation details to Appx. B. Below are the four IL baselines we use to compare with SLIL:

- *Behavior Cloning (BC)* (Pomerleau, 1991): Expert demonstrations as a set of state-action pairs are split into 70% training data and 30% validation data. The learner policy is trained with supervised learning where actions are viewed as labels and states are viewed as input features.
- *Generative Adversarial Imitation Learning (GAIL)* (Ho & Ermon, 2016): GAIL is an IL method that consists of a generator as a policy network mimicking the expert behaviors, and a discriminator as a reward signal distinguishing between learner and expert behaviors.
- *Generative PRedecessor models for Imitation Learning (GPRIL)* (Schroecker et al., 2019): GPRIL performs the state-action distribution matching by jointly training the learner policy and the corresponding multi-step predecessor state-action distribution. In each iteration, the predecessor state-action distribution is estimated using masked autoregressive flows (Papamakarios et al., 2017).
- *Disagreement-Regularised Imitation Learning (DRIL)* (Brantley et al., 2019): DRIL pre-trains an ensemble of BC policies with expert demonstration data, and uses RL to train a learner policy whose cost function is proportional to the sum of the variance of ensemble policies' predictions.

### 5.1   MODE COVERAGE

Fig. 1 shows the results of Reacher4 with four mode targets in different colors in Fig. 1a, and Fig. 7 shows the results of HalfCheetah2 with two mode directions as running forward and backward.

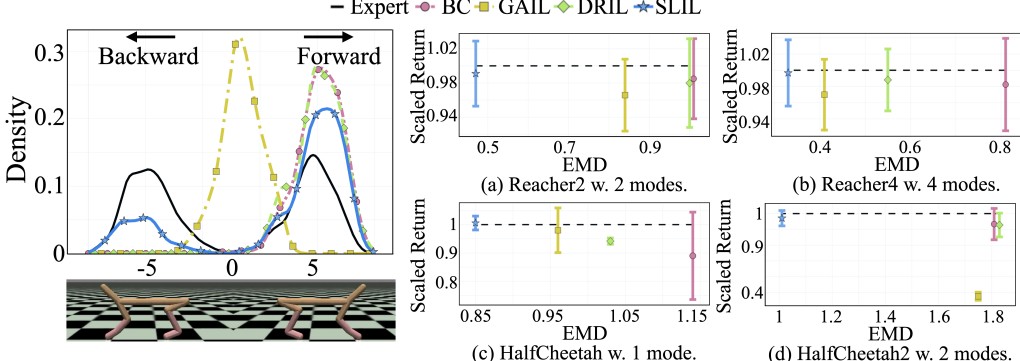

Figure 7: Results of SLIL (Ours) and baselines on mode coverage in HalfCheetah2. All the distributions are visualized using KDE (Sheather & Jones, 1991).

Figure 8: EMD vs scaled return in tasks with multiple modes. The x-axis is the EMD (Ling & Okada, 2007) between expert and learner policy state distribution. The y-axis is the expected return (*i.e*, total reward), scaled so that the expert achieves 1 and a random policy achieves 0.

Results of Reacher2 are in Appx. C as its observations are similar to the Reacher4 task. Fig. 1b shows the mode coverage (*i.e*, the state distribution) of the expert policy, where the expert tends to cover all four targets (as four modes) evenly. The learner policy obtained by SLIL preserves the expert mode coverage very well as shown in Fig. 1c. On the other hand, the learner policy from GAIL is prone to mode collapse. It only focuses on the green target out of the four targets. Consistent with results in SOTA works (Fei et al., 2020; Arjovsky et al., 2017; Jena & Sycara, 2020), this mode collapse is due to the adversarial training process used in GAIL. DRIL also fails to cover the green target mode as its defined reward function likely encourages a mode seeking behavior. SLIL successfully preserves the mode coverage from the expert, because it uses our DCNF to accurately estimate the expert state distribution, and the stabilized two-stage training to update the learner policy.

In the HalfCheetah2 task shown in Fig. 7, the x-axis represents the running velocity, and the plots show the velocity distributions of expert and IL policies. The black curve demonstrates two modes (*i.e*, running forward and backward) in expert demonstrations. SLIL (blue curve) is able to preserve all modes, while DRIL and BC are collapsed to running forward and GAIL fails to reveal any mode. We further calculated the earth mover's distance (EMD) (Ling & Okada, 2007) between expert and learned policies' state distributions as the x-axis in Fig. 8 in the Reacher and HalfCheetah tasks with multiple modes. A lower EMD value indicates a better learner policy at recovering expert demonstration modes, and the results echo the above observations quantitatively. We omit GPRIL as no meaningful results are obtained.

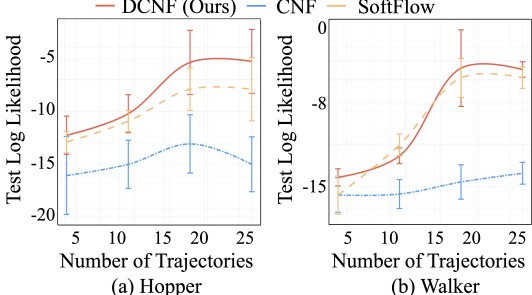

Figure 9: Test state log likelihood using $P_{E_{\omega^*}}$ learned from our DCNF and CNF.

| # of | Hopper | | | Walker | | |
|---|---|---|---|---|---|---|
| Traj. | CNF | SoftFlow | DCNF | CNF | SoftFlow | DCNF |
| 4 | 3.73 | 1.88 | **0.63** | 3.22 | 2.04 | **1.44** |
| 11 | 2.69 | 1.42 | **0.52** | 3.72 | 1.53 | **0.80** |
| 18 | 2.09 | 0.94 | **0.49** | 3.73 | 1.10 | **0.80** |
| 25 | 1.95 | 0.68 | **0.49** | 2.85 | 1.02 | **0.64** |

Table 1: The EMD (Ling & Okada, 2007) between the Gaussian and the latent noise recovered from $P_{E_{\omega^*}}$ learned from DCNF, SoftFlow and CNF.

## 5.2 EXPERT STATE ESTIMATION

To show how our proposed denoising mechanism in DCNF addresses the manifold hypothesis challenge in estimating the expert state distribution, we compare our DCNF to SoftFlow and CNF both in terms of i) *expert state matching* and ii) *latent space (*i.e*, Gaussian noise) recovery*. The results below are obtained using Hopper and Walker tasks. We make similar observations for other tasks, so their results are omitted for brevity.

*i) Expert state matching.* To quantify the quality of the expert state matching, we use expert policies to generate 50 trajectories (each consisting of 1,000 state-action pairs) as a test set and evaluate the log-likelihood of states in the test set using $P_{E_{\omega^*}}$ learned by our DCNF, SoftFlow (Kim et al., 2020a)

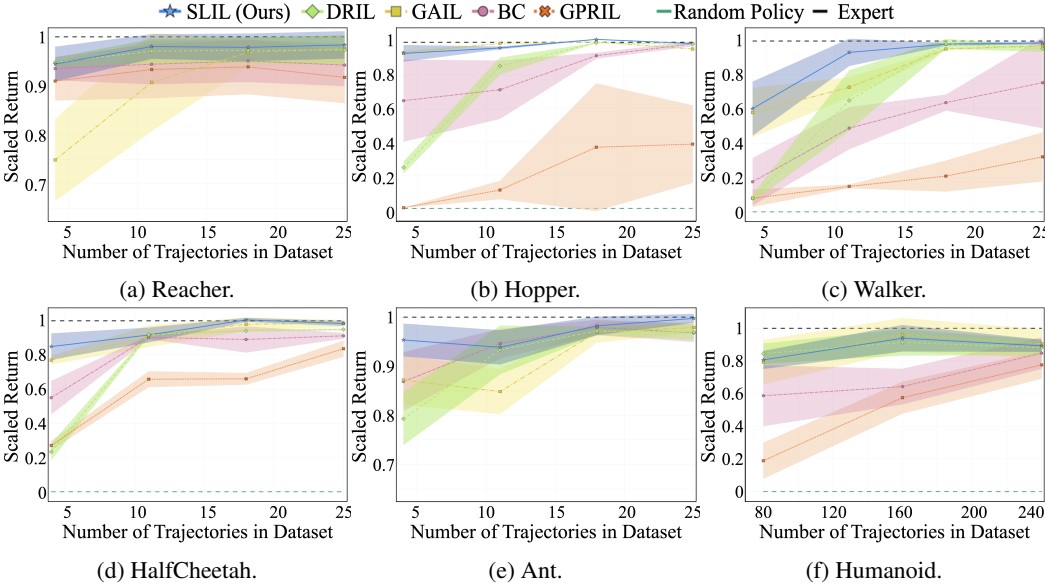

Figure 10: Performance of learner policies in tasks with one mode. The y-axis is the expected return (*i.e*, total reward), scaled so that the expert achieves 1 and a random policy achieves 0.

and CNF (Grathwohl et al., 2018). A higher test state log-likelihood score indicates a more accurate model in estimating the expert state distribution. The results are shown in Fig. 9. Clearly, in both tasks with different numbers of expert demonstrations, DCNF outperforms SoftFlow and CNF which validates the necessity of our denoising mechanism in tackling the manifold hypothesis challenge for expert state distribution estimation.

*ii) Latent space (i.e., Gaussian noise) recovery*. We use the test set to recover latent noises and compute its EMD (Ling & Okada, 2007) to the Gaussian noise distribution. A lower EMD value indicates a better normalizing flow model, with its learned data transformation closer to a homeomorphism. Tab. 1 shows the EMD results in Hopper and Walker with different number of expert demonstrations. It demonstrates that DCNF is more likely to recover a latent space closer to Gaussian.

## 5.3 PERFORMANCE OF THE LEARNER POLICY FROM SLIL

Fig. 8 shows the EMD (Ling & Okada, 2007) vs scaled return results in Reacher and HalfCheetah with different numbers of demonstration modes. In all the tasks, SLIL has both lower EMD and higher scaled return compared with baselines. This is because DCNF is able to recover expert state distribution $P_E$ with multiple modes well, and thus provides useful feedback for the learner policy to recover expert behaviors. Comparing between Reacher with 2 and 4 targets, SLIL shows a larger return margin when the target modes are more distant from each other in the Reacher2 task.

Fig. 10 shows the performances of the learner policies from our SLIL and IL baselines under different numbers of expert trajectories when they only contain one mode. In all tasks, the learner policies from our SLIL have comparable performances with GAIL, which is because SLIL directly maximizes the log-likelihood of expert data without using adversarial training. SLIL outperforms DRIL particularly with a limited number of expert trajectories because the use of DCNF-learned $P_E$ has good generalization ability and provides useful feedback to learner policy $\pi$ to reach a better policy. Moreover, in both easy tasks (Reacher) and complex tasks (Hopper and Walker), SLIL consistently outperforms BC with different numbers of expert demonstrations, which is because SLIL uses the state distribution matching on top of the BC objective to overcome the covariate shift problem. The performance of the learner policy from BC increases when using more expert demonstrations, as more training data mitigate the overfitting problem and compounding errors. However, the learner policies from GPRIL have the lowest performances in all tasks. This is primarily because the policy is jointly learned with its multi-step predecessor state-action distribution. With random initial parameters for these two functions, it is hard to progressively improve them jointly.

## 5.4 TRAINING STABILITY OF SLIL

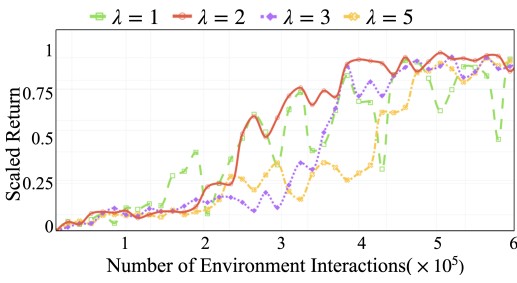

Figure 11: SLIL (Ours) performance with different gating parameters $\lambda$ in Walker.

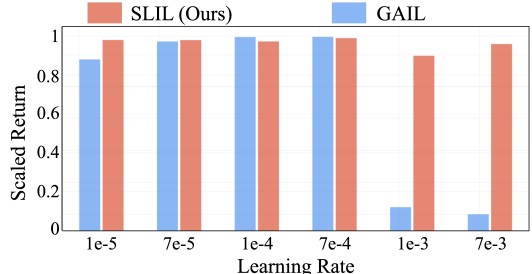

Figure 12: Policy performances with different learning rates using GAIL and our SLIL in Walker.

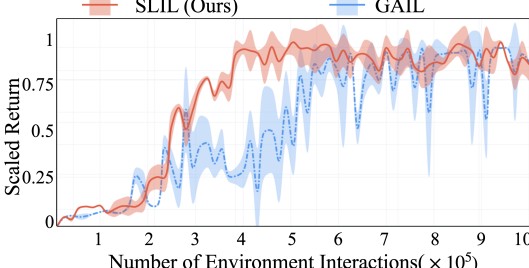

Figure 13: Policy performance over environment interaction numbers (SLIL vs GAIL) in Walker.

When training the learner policy, the convergence speed, performance fluctuation, and robustness to the learning rate are all important. A *stable IL algorithm* leads to a high convergence speed, low performance fluctuation in training, and works in a wide-range of learning rate choices. Now, we investigate the training stability of our SLIL from the perspectives of i) the gating parameter $\lambda$, ii) the learning rate, and iii) the number of environment interactions. Below, we show our results from the Walker task with 18 expert trajectories; similar observations were made for other tasks.

*Impact of the gating parameter $\lambda$.* Fig. 11 shows the training curves of SLIL with different gating parameters $\lambda$. It is clear that when $\lambda = 2$, the SLIL training process is stable, with small performance fluctuation, and fast convergence rate. On the other hand, a lower gating parameter (*i.e*, $\lambda = 1$) leads to an unstable training curve, with high fluctuations; and a higher gating parameter e.g., $\lambda = 3$ and $\lambda = 5$ leads to a slower convergence rate. This indicates that both behavior cloning and expert state matching matter in learning expert policy, and balancing the weights of these two components is crucial. If the weight to the expert state matching is too much (small $\lambda$), it leads to a high training fluctuation. On the other hand, when the weight of the expert state matching is too small (large $\lambda$), it slows down the training speed.

*Impact of the learning rate.* We further study SLIL's robustness in choosing different learning rates. Fig. 12 shows the performance of SLIL and GAIL given different learning rates. It shows that SLIL works in a wide range of learning rates, while GAIL tends to fail/crash when learning rates are 1e-3 and higher. GAIL with adversarial training is less robust to the change of learning rate due to vanishing gradient and the complex interactions between the discriminator and the generator (Jena & Sycara, 2020; Arjovsky et al., 2017). However, our SLIL is implemented in a two-stage optimization framework, which is more robust to hyper-parameter changes.

*Impact of the number of environment interactions.* Fig. 13 shows the performance change with the number of environment interactions (self-supervision steps). This figure shows that SLIL is able to attain a higher return than GAIL given a small number of environment interactions (at around 3–6×$10^5$ steps). An explanation is that the pre-trained $P_{E\omega^*}$ is good at guiding the learner policy $\pi_\theta$ to explore on those states frequently visited by the experts. Moreover, differing from the adversarial training in GAIL with two generator and discriminator trained alternatively, SLIL employs the two-stage training process to training $P_{E\omega^*}$ and $\pi_\theta$ separately, thus leading to a fast convergence rate.

## 6 CONCLUSION

In this work, we proposed SLIL – stable likelihood-based imitation learning – which trains a learner policy by directly maximizing the likelihood of expert demonstrations. SLIL is a stable two-stage optimization framework, where in stage one we accurately estimate the expert state distribution using a novel denoising continuous normalizing flow method, and in stage two we train the learner policy to match both expert's policy and state distribution. Comparing our SLIL with baselines in ten different physics-based control tasks, we present superior evaluation results in terms of learner policy performance, training stability, and mode distribution preservation.

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
