# OpenReview forum: "Stabilized Likelihood-based Imitation Learning via Denoising Continuous Normalizing Flow"
_ICLR.cc/2022/Conference — ICLR 2022 Submitted_

### Official Review · Reviewer_SsHp · 2021-10-21

**Correctness:** 2
**Technical Novelty And Significance:** 2
**Empirical Novelty And Significance:** 2
**Recommendation:** 5
**Confidence:** 5

**Main Review:**

Strong Points
* The empirical results are good: SLIL achieves good results, no matter whether few demonstrations or many demonstrations are provided.
* The presentation is good, however, I have some suggestions below.

Weak Points
* Technical correctness (1): The main claim (Theorem 4.1) that SLIL tightly bounds the likelihood objective (LIL) is not supported. The SLIL objective differs by substituting the term $E_{p_\text{E}(s)}\big[ \log p_{\pi}(s) \big]$ with $E_{p_{\pi}(s)}\big[ \log p_{\text{E}}(s) \big]$, that is, switching the expert' and agent's state distribution. The proof has several issues: First of all, it starts with circular reasoning, by assuming that the optimal policy of the SLIL objective, $\pi^\star$,  matches the expert's state distribution. It then derives the bound (which is only valid for $\pi^\star$) that uses the equality $\max_x  f(x) + \max_x g(x) \ge \max_x \big[ f(x) + g(x) \big]$ which is not at all tight in general.
* Technical correctness (2): The SLIL objective is further modified by introducing a hyper-parameter that disables the RL objective ever $\lambda$ iterations, which effectively changes the objective being optimized without any theoretical justification.

* Novelty (1): SLIL is a hybrid between BC and Energy-Based Imitation Learning (EBIL) [2], a reference certainly missing. Like SLIL, State-only EBIL starts by estimating the expert's state distribution and uses it as a fixed reward function for imitation learning. As the SLIL loss is basically an interpolation between two prior imitation learning losses the contribution seems minor, in particular given that the objective of SLIL is not well motivated (Technical correctness (1)). It is also worth noting that AIRL uses noise contrastive estimation to estimate the expert policy and uses this density estimate directly as reward for MaxEnt-RL, which is similar to the 2nd term of the SLIL objective. The differences are thus that AIRL ues MaxEnt-RL instead of RL, $r=\log p_E(a|s)$ instead of $r=\log p_E(s)$, and NCE with an EBM for density estimation rather than maximum likelihood with normalizing flows.

* Novelty (2): The "Denoising continuous normalizing flow" seems to be a minor variant of Softflow [Kim et al. 2020]. Furthermore, this minor contribution is not evaluated at all (neither Softflow, nor any other methods for handling the manifold hypothesis challenge were compared).

* Evaluation:  The improved stability compared to GAIL was not demonstrated. Please add learning curves for the data of Fig. 10 to the Appendix (choosing a suitable number of demonstrations (e..g. 5) for each experiment).

Additional Comments
* It is not clear why the hyper-parameter lambda was introduced in favor of a fixed coefficient that trades off the two objectives, which would at least maintain a fixed objective function.
* Much of 4.2 seems to be related work. I think it would make more sense to put it before the "Method" section to make the contributions more clear. In exchange, the problem formulation of LIL could be put under method, since it is introduced to "derive" the SLIL objective.
* Algorithmbox 1 is almost useless, because it mainly shows that we have an iterative training loop, but doesn't really show what happens at each iteration.
* End of page 5. The definition of $P_{E,\sigma}(\tilde{s}|s)$ looks like  $P_{E,\sigma}(\tilde{s},s)$ to me.

References:
[1] Liu, R., Gao, J., Zhang, J., Meng, D., & Lin, Z. (2021). Investigating bi-level optimization for learning and vision from a unified perspective: A survey and beyond. arXiv preprint arXiv:2101.11517.

[2] Liu, M., He, T., Xu, M., & Zhang, W. (2020). Energy-based imitation learning. arXiv preprint arXiv:2004.09395.

**Summary Of The Paper:**

The paper present a novel method for imitation learning aiming to stabilize training and mode coverage compared to existing methods. The proposed method (SLIL) extends behavioral cloning by an additional objective of maximizing a state-only reward that is defined by a density estimate (trained once beforehand) of the expert's state distribution.
For density estimation SLIL uses a continuous normalizing flow where the expert states are corrupted by Gaussian noise, where the noise-level is decreased during training. The noise is introduced to tackle the "Manifold Hypothesis Challenge", that is, the inability of normalizing flows to accurate approximate distributions that lie on a lower-dimensional manifold.

The main contributions are:
1. A novel imitation learning method that combines the BC loss with a RL loss for reaching expert states
2. A slightly modified approach to tackle the manifold hypothesis challenge (Kim et al. [2020] used Gaussian noise where the noise-level was sampled from a uniform distribution, independently for each data set, rather than using the same noise-level for all data and annealing it during optimization)

**Summary Of The Review:**

The contributions seem rather incremental and the claims are not well-supported.
I still gave a relatively high recommendation, because the results are fine and the overall ideas of 1) tackling covariance shift of BC by using an RL objective to match the expert state distribution, and 2) estimating the expert state distribution once instead of iteratively based on NCE with adapted noise to improve stability, are reasonable.

---

> ### Author Response · Authors · 2021-11-19
> **Response to Reviewer SsHp [Part 1/2]**
>
> We thank the reviewer for the careful review and constructive suggestions for the paper. Given the reviews, we have addressed most questions in the updated revision. Due to limited space, we committed two comments to respond.
>
> #### **W1:** "Technical correctness (1): The main claim (Theorem 4.1) that SLIL tightly bounds the likelihood objective (LIL) is not supported. ... First of all, it starts with circular reasoning, by assuming that the optimal policy of the SLIL objective, $\pi^*$, matches the expert's state distribution. It then derives the bound (which is only valid for $\pi^*$) that uses the equality $\max_x f(x)+\max_x g(x) \geq \max_x[f(x)+g(x)]$.''
>
> **R1:** Thm. 4.1 shows that the LIL objective in Eq.(2) aims at learning a policy that matches expert state and action distributions, and it can be lower bounded by the SLIL problem in Eq.(3).
>
> - We do not understand what is circular reasoning and why it exists in the proof, and are wondering if the reviewer can give more explanations about it.
>
> - We agree with the reviewer that in general $\max_x f(x)+\max_x g(x) \geq \max_x[f(x)+g(x)]$ is not tight. However, considering that in Thm. 4.1, $f(x)$ and $g(x)$ represent $\mathbb{E}\_{(s,a)\in\mathcal{D\}\_E}[\log \pi(a|s)]$ and $\mathbb{E\}_{s\in\mathcal{D}\_\pi}[\log P\_E(s)] $ respectively which are all non-positive, equality can be reached and the inequality is tight.
>
> #### **W2** "Technical correctness (2): The SLIL objective is further modified by introducing a hyper-parameter that disables the RL objective every $\lambda$ iterations, which effectively changes the objective being optimized without any theoretical justification."
>
>
> **R2:** Introducing $\lambda$ to disable RL objective in some training iterations is a practical trick that can be applied to both improve sample efficiency (decrease the number of interactions with the environment) and training stability (as is shown in Fig. 11). Such a trick can be viewed as a strategy of learning rate control, e.g., by Lee., et al., Deeptwist, arXiv, 2019, and thus does not conflict with the SLIL objective and training scheme. In addition, Fig. 11 shows that without using any gating mechanism (i.e., $\lambda=1$), SLIL still shows good performance in the Walker task.
>
>  #### **W3** "Novelty (1): SLIL is a hybrid between BC and Energy-Based Imitation Learning (EBIL), ... As the SLIL loss is basically an interpolation between two prior imitation learning losses the contribution seems minor, in particular given that the objective of SLIL is not well motivated (Technical correctness (1)). It is also worth noting that AIRL uses noise contrastive estimation to estimate the expert policy and uses this density estimate directly as reward for MaxEnt-RL, which is similar to the 2nd term of the SLIL objective. ..., and NCE with an EBM for density estimation rather than maximum likelihood with normalizing flows. "
>
> **R3:** We thank the reviewer for pointing out the interesting reference - EBIL. However, our proposed SLIL is not a combination of EBIL and BC, and BC+EBIL cannot address the mode collapse and training stability problems. See details below:
>
> - **Problem definition:** Our proposed SLIL targets at solving the mode collapse and training instability problems in GAIL. EBIL targets at avoiding adversarial training instead.
>
> - **Technical contribution:** In SLIL in Eq.(3), we use the lower-level optimization result as well as the second term in the upper-level optimization objective to help the learner policy match expert state distribution to address to mode collapse problem. In contrast, EBIL targets at minimizing the reverse KL divergence between expert and learner state-action occupancy measures which encourages a mode-seeking behavior (Ghasemipour., et al, arXiv:1911.02256, 2019), which does not match the state distribution of the learner $\pi$ to the expert $\pi_E$, thus fails to address the mode collapse problem.
>
> As a result, our proposed SLIL is significantly different from BC + EBIL. We cited EBIL as a related work in the updated revision. We do not compare it as a baseline since the released code is incomplete to compile and the limited time.

---

> > ### Author Response · Authors · 2021-11-19
> > **Response to Reviewer SsHp [Part 2/2]**
> >
> > #### **W4:** "Novelty (2): The "Denoising continuous normalizing flow" seems to be a minor variant of Softflow [Kim et al. 2020]. Furthermore, this minor contribution is not evaluated at all ..."
> >
> > **R4:**: DCNF is not a minor variant of SoftFlow for the following reasons:
> > - **Technical difference:** Different from SoftFlow which uniformly samples Gaussian noise levels to perturb data, DCNF decreases Gaussian noise levels in each training iterations so that it can focus more on learning expert state distributions while overcoming the manifold hypothesis.
> >
> > - **Experiment support:**
> >     1. In the updated revision, we added the performance of SLIL implemented with SoftFlow to model $P_E$ on the Reacher4, Hopper and Walker tasks shown in Fig. 1 (page 2), Fig. 9 and Tab. 1 (on page 8) respectively. The experiment results show that DCNF is able to recover expert state distributions more precisely in the Hopper and Walker tasks than SoftFlow, and outperforms it in recovering expert modes in the Reacher4 task.
> >
> >     2. Our hypothesis of the positive effects of the denoising mechanism in DCNF is that it is able to make the loss landscape smoother during training than SoftFlow. We have shown the learning curves of SoftFlow and DCNF in Fig. 14 in the Hopper task in the updated Appx. C. It shows that DCNF has lower training loss (in expert state NLL) than SoftFlow in each epoch. This likely indicates that the denoising mechanism is able to find better local minima by imposing different levels of regularizations in each training epoch.
> >
> > In the updated revision, we have edited the DCNF algorithm in Alg. 1 on page 6 to further clarify and highlight the difference between DCNF and SoftFlow.
> >
> > #### **W5:** "Evaluation: The improved stability compared to GAIL was not demonstrated. Please add learning curves for the data of Fig. 10 to the Appendix (choosing a suitable number of demonstrations (e..g. 5) for each experiment)."
> >
> > **R5:** We added the learning curves in the Appx C in Fig. 16. In general, in all tasks, the SLIL approach has higher convergence speed with smoother learning curves than GAIL.
> >
> > #### **W6:**  "It is not clear why the hyper-parameter lambda was introduced in favor of a fixed coefficient that trades off the two objectives, which would at least maintain a fixed objective function."
> >
> > **R6:** Please refer to **R2** above for details. $\lambda$ can be treated as a learning rate policy control trick in practical SLIL training.
> >
> >  #### **W7:** "Much of 4.2 seems to be related work. I think it would make more sense to put it before the "Method" section to make the contributions more clear. In exchange, the problem formulation of LIL could be put under method, since it is introduced to "derive" the SLIL objective."
> >
> >  **R7:** Thank you for your suggestion, we will make this modification in the final version of the paper.
> >
> > #### **W8:** "Algorithmbox 1 is almost useless, because it mainly shows that we have an iterative training loop, but doesn't really show what happens at each iteration."
> >
> > **R8:** We deleted it in the updated revision following your suggestion.
> >
> > #### **W9:** "End of page 5. Joint probability problem."
> >
> > **R9:** To make it more clear, we updated the statement at the bottom of Page 5 to:
> >
> > Denote a perturbed expert state $s\sim\mathcal{D}_E$ as $\tilde{s}$, with $\tilde{s}=s+\sigma \epsilon$
> > where $\epsilon\sim \mathcal{N}(0; I)$. When the noise level is very small, i.e, $\sigma \rightarrow 0$, $\tilde{s}$ approaches $s$.

---

> > ### Comment · Reviewer_SsHp · 2021-11-22
> > **Regarding Technical Soundness**
> >
> > Thank you very much for your reply and updates to the revision, and apologies for my last-minute reply.
> >
> > 1. By circular reasoning, I mean that Theorem 4.1 is proven based on the assumption that the optimal policy will match the expert distribution, which would only be the case if Theorem 4.1 was true. However, one could also prove that $0=1$ based on the assumption that $0=1$, which is clearly not useful.
> >
> > 2. The fact that f and g are non-positive does not imply that equality holds, or that the bound is tight. There could be $x_1 != x_2$ such that $f(x_1)$ and $g(x_2)$ are both very large, but still there might not be a single x such that both terms are large, that is $max_x f(x) + g(x)$ can be significantly smaller than $max_x f(x) + max_y g(y)$.
> >
> > Regarding EBIL+BC hybrid:
> > 1. It is not clear to my why minimizing the RKL does not match the expert distribution. Minimizing any divergence will at its optimum match the expert distribution (when assuming that this is possible).
> >
> > 2. I agree that the motivation for EBIL and SLIL are different, and I also agree that the methods are different. Yet, SLIL seems to use a combination of BC (the first term of Eq. 3) and the EBIL objective (the second term of Eq. 3), by summing both objectives.

---

> > > ### Author Response · Authors · 2021-11-22
> > > **Thank you for your reply and comments!**
> > >
> > > Thank you so much for explaining your doubts before the rebuttal window is closed. We really appreciate it. Please find our answers below.
> > >
> > > **1. Circular reasoning.** First of all, please double check our statement in the theorem that *we are never claiming that the learned policy will be exactly the same as the optimal policy if Eq 3 holds.* Secondly, we do not assume that $\pi = \pi^*$ is the exact and only solution so that the equation in our proof holds. Instead, we state that "where equality holds when $\pi=\pi^*$, AT LEAST". The assumption we made here is that the optimal policy, as one desirable solution, should achieve a global minimum of our objective. Therefore, we do not use any circular reasoning at all in our proof.
> > >
> > > **2. Tightness.** Generally speaking, you are correct. However, in a special case where f and g achieve their global maximum at the same point, it will be true that $\max_x (f(x)+g(x)) = \max_x f(x) + \max_x g(x)$. Our proof is based on this special case, where when $\pi=\pi^*$ holds, both $D_E = D_{\pi} = D_{\pi^*}$ and $P_E = P_{\pi} = P_{\pi^*}$ holds as well, leading to a global maximum of each term with equality.
> > >
> > > **3. Why RKL does not matching expert distribution???** Ideally, as the ultimate goal, any imitation learning algorithm should be able to match the expert distribution, including RKL. However, the key question is "HOW". At least, in [Zhang et al. f-GAIL, NeurIPS 2020], we can observe that simply minimizing a divergence objective cannot match the expert distribution. The reviewer's assumption of "when assuming that this is possible" is probably impossible as far as we know in the literature. What is the expert policy? Is it unique? How can it be recovered for matching? What/How can matching techniques be successful in matching? How to avoid critical points in optimizing the objective function? Any theoretical guarantee? etc. There are too many open questions related to the reviewer’s question that we do not have answers yet, but we are aiming to do so.
> > >
> > > **4. Differences between EBIL and the second term of Eq. 3.** It is very inappropriate to take both as being the same by just simply looking at their formulas (they are even different in the formulas). They are significantly different in, at least, the following three aspects:
> > >
> > > *(1) Modeling assumptions:* In EBIL, there is a clear Gaussian assumption in the probability based on the energy (see Eq. 7, 9, 10, 11, 12, 14, 15 in [Liu et al. AAMAS 2021]). In contrast, there is NO assumption on our second term as probability. That is, our probability is purely data-driven so that we hope to better recover the "optimal" policy with no mode collapse. In this sense, we may think that our second term may be more general than EBIL.
> > >
> > > *(2) Formulas:* EBIL is an unconstrained optimization problem, while ours is a clear constrained optimization problem. Therefore, it will be much more challenging to solve our problem in the context of imitation learning.
> > >
> > > *(3) Proposed solutions:* EBIL is based on KL divergence to solve their problem. In contrast, we propose a novel denoising continuous normalizing flow (DCNF) based approach as our solver.
> > >
> > > As a consequence, our approach leads to a very different algorithm from EBIL as well. We will add such discussions in the paper.

---

> ### Author Response · Authors · 2021-11-28
> **Has our response addressed your concerns?**
>
> Dear reviewer SsHp,
>
> we would be grateful if you can confirm whether our response has addressed your concerns, and let us know if any issues remain. Please take a look at our response, the paper and appendix of the revised version for further details.

---

> > ### Comment · Reviewer_SsHp · 2021-11-29
> > **Not Sufficiently**
> >
> > Dear authors,
> >
> > I do not think that my concerns have been sufficiently well addressed to increase my score.
> >
> > 1) The claim that Eq.2 is tightly lower bounded by Eq. 3 is wrong. The proof of this statement (Theorem 4.1) states that you assume that $\pi = \pi^\star$, which is already a strong assumption and furthermore implies that the "tight bound" only happens at the optimum and thus only shows that both objectives share this stationary point. Furthermore to be more precise, you actually assume that $\pi^\star$ matches the expert distribution perfectly, which is an even stronger and completely unrealistic assumption because the parametric policy is in general not possible to match the empirical expert distribution perfectly (as you admitted yourself in your previous reply). The proof of Theorem 4.1 could be used with slight modifications to prove that any divergence-minimizing imitation learning methods are equivalent. I literally do not find any value in this proof, quite contrary it is rather misleading. It would seem more sensible to directly motivate the objective of Eq. 3, rather than trying to derive it from Eq. 2.
> >
> > 2) I think not only the technical contribution (the provided "proofs") but also the methodological contribution is marginal, because (on a high level) you merely add a BC loss to a known objective (using a direct density estimate of $P_E$ as reward). I do see that there are several further minor variations down the road (in particular how you estimate the expert density using your "denoising" normalizing flows, and by adding a stepsize-hyperparameter to updating one of the terms only every x iterations), but these modifications have little theoretical justifcations (if at all) and improve the performance at the cost of introducing more hyperparameters.
> >
> > I think that showing modest empirical improvements with additional hyperparameters on your  benchmark is not sufficient to justify publication, given the absence of a convincing theoretical contribution.

---

> > > ### Author Response · Authors · 2021-11-29
> > > **Update**
> > >
> > > Thank Reviewer SsHp for your prompt response.
> > >
> > > 1. We didn't claim an assumption of $\pi=\pi^*$ in Theorem 4.1. So there is no circular reasoning problem. By using the samples of the expert and the parametric policies, it is of course not possible to match exactly the two policies. We theoretically prove that it holds by expectation with $\mathbb{E}[\cdot]$. Moreover, Theorem 4.1 is not equivalent to other divergence-minimizing imitation learning methods, because the objectives are totally different (See [F1] for how different divergence-minimizing methods vary, as we showed in our previous response). Moreover, the works starts from eq 1, a more explainable motivation/objective with a clear physical meaning, say, to learn the policy $\pi$ by directly maximizing the likelihood of the state-action pairs from the expert demonstration data. If starting from eq 3 (even thought it works), the solution would seem arbitrary, and not well explained/reasoned.
> > >
> > > [F1] Zhang, Xin, et al. "f-GAIL: Learning $ f $-Divergence for Generative Adversarial Imitation Learning." NeurIPS 2020 (arXiv preprint arXiv:2010.01207 (2020)).
> > >
> > > 2. Our approach is novel in addressing the mode-collapse problem of imitation learning. As you mentioned, though there have been many publications, like EBIL, with the objective as BC plus a second term. However, the second term in our SLIL is uniquely designed to tackle the mode-collapse problem, while no other works (including EBIL) can solve. All other approaches, such as EBIL is not capable of solving the mode-collapse. As we highlighted in our previous response, our technical contributions lie in three aspects, including 1) the novel objective design particularly for mode-collapse problem, 2) the objective is clearly a constrained optimization, differing from EBIL, 3) a novel denoising continuous normalizing flow (DCNF) based approach as our solver.

---

> > > > ### Comment · Reviewer_SsHp · 2021-11-29
> > > > **Please clarify**
> > > >
> > > > 1. The proof of Theorem 4.1 contains a line $\overset{\pi=\pi^\star}{=}$ which I read as: "the equality is only true under the assumption that the current policy $\pi$ is optimal (in the sence that $P_{\pi^\star} = P_E$)". Hence, the bound is only proven to be tight when the current policy matches the expert distribution correctly. Under the same assumption, one can straightforwardly prove that any divergence-objective (A) is a tight bound of any other divergence-objective (B), as they will all equal to $0$. However, these bounds are only tight when the current policy already matches the expert perfectly and are, thus, hardly useful.
> > > >
> > > > 2. Please clarify how your "constrained" optimization problem in Eq. 3, differs from setting $P_E$ to the MLE before solving the unconstrained objective.

---

> > > > > ### Author Response · Authors · 2021-11-29
> > > > > **Clarification**
> > > > >
> > > > > We really appreciate your responses. They made your concerns clear to us, which will significantly improve the quality of our work.
> > > > >
> > > > > 1. Thanks for clarifying your circular reasoning concern. In fact, the equation with $\overset{\pi=\pi^\star}{=}$ shows the fact when ${\pi=\pi^\star}$, where ${\pi=\pi^\star}$ is not the assumption. It means that ${\pi=\pi^\star}$ leads to the maximum of our objective, but there might be multiple $\pi$'s that achieve the same maximum, due to the non-convexity of our objective. Therefore, ${\pi=\pi^\star}$ is one of the possible solutions to maximize our objective, not an assumption. **The only assumption we have here is that $\pi^\star$ can maximize our objective, which commonly exists in imitation learning literature, such as GAIL, etc.**
> > > > >
> > > > > Moreover, in terms of the divergence-objectives, e.g., A and B. It is true that A=B=0, when $\pi=\pi^\star$, but there is no guarantee of a bound, e.g., $A\geq B$ or $A\leq B$, when $\pi \neq \pi^\star$. As a result, there is no need to discuss the tightness of bounds.
> > > > >
> > > > >
> > > > > 2. The unique **constrained** optimization problem is referred to Eq 1 and Eq 2 (not Eq 3), which defines our original LIL problem as a bilevel optimization problem. However, the constrained problem is challenging to solve. In Theorem 4.1, we bound it by eq 3, with $P_E$ as a parameter to be learned from $D_E$. Hence, we relax Eq 1 and Eq 2 to Eq 3 as an unconstrained problem.
> > > > >
> > > > > We really appreciate your insightful comments. We hope that our response addresses your concerns successfully.

---

> > > > > > ### Comment · Reviewer_SsHp · 2021-11-30
> > > > > > **My concerns are not addressed**
> > > > > >
> > > > > > 1. The problem with the proof and consequently Theorem 4.1 is that it only considers the original objective at a single point $\pi = \pi^\star$. When lower-bounding a function, one should proof that
> > > > > >
> > > > > >  $\forall x: f(x) \le g(x)$,
> > > > > >
> > > > > >  and not
> > > > > >
> > > > > > $\forall x: f(x) \le g(x^\star)$.
> > > > > >
> > > > > > The latter bound does not tell you anything about g(x), and thus for any $x\neq x^\star$, g(x) might be arbitrarily bad, even if f(x) is good. In your case, this means that for any policy that does not match the expert distribution perfectly, you have no guarantees about the quality on the original objective. An analogous proof can be constructed for any two divergences $D_a(\pi||\pi_E)$ and $D_b(\pi||\pi_E)$:
> > > > > >
> > > > > > $
> > > > > > \underset{\pi}{max} -D_a(\pi||\pi_E) \\
> > > > > > \overset{\pi = \pi^\star}{=} -D_a(\pi^\star||\pi_E) \\
> > > > > > = 0 \\
> > > > > > \le \underset{\pi}{max} -D_b(\pi||\pi_E)
> > > > > > $
> > > > > >
> > > > > > You argue that there is no guarantee for a bound $D_b(\pi||\pi_E) \le D_a(\pi||\pi_E)$, when $\pi \neq \pi^\star$, and I agree. This is exactly what I criticize about your proof: Where do you prove the bound for general $\pi$?
> > > > > >
> > > > > > 2.  SLIL optimizes Eq.3 which only differs by the additional BC term (the method itself also differs in the way the MLE is computed). I agree that you present a derivations based on a different formulation, but a) theis derivation is not convincing (see 1.) and b) this does not affect my critique that the proposed _method_ maximizes the sum of two known objectives, which is a marginal contribution.

---

### Official Review · Reviewer_nLuT · 2021-10-27

**Correctness:** 3
**Technical Novelty And Significance:** 3
**Empirical Novelty And Significance:** 3
**Recommendation:** 8
**Confidence:** 4

**Main Review:**

Main strength
--
This paper's main strength is that it proposes a straightforward method that both conceptually and empirically satisfies important desiderata of imitation learning methods (1) address the cascading error problem of BC-based methods (2) address the instability (mode collapse) of adversarially-trained IL methods (3) avoid dependence on an interactive demonstrator (i.e. DAGGER).

Main weakness
--
The paper's main weakness is some imprecision in a few places

### Imprecision
There are some issues (a,b,c) with the proof. (a) The proof refers to component $\enclose{circle}{2}$ in Eq 1, yet Eq. 1 has no denoted components. Is this reference correct? If so, then Eq 1. should denote the second component with an underbrace. If not, should it instead be Eq. 2 or Eq. 3? (b) The proof states "with sufficient expert data $\mathcal D_E$, $\enclose{circle}{2}$ can be expressed with $\sum_{s \in \mathcal D_\pi} \frac{P_E(s)}{P_\pi(s)} \log P_\pi(s)$. I don't see how this is true, because $\enclose{circle}{2}$ in Eq (3) is $\sum_{s \in \mathcal D_\pi} \log P_E(s)$, i.e. the state distribution is the state distribution of the policy. Again, is the reference to $\enclose{circle}{2}$ correct? These ambiguities / errors prevent me from being able to completely follow the proof. (c) It's not obvious from the definition of $g(\pi, P_\pi)$ that $\sum_{s \in \mathcal D_{\pi^*}} \log P_{\pi^*}(s) \geq \max_\pi \sum_{s \in \mathcal D_\pi} \log P_E(s)$ for two reasons (c.i) because the sums are not normalized by the the dataset size, it would be easy to invalidate the inequality by changing the dataset sizes (c.ii) What if $\pi$ was such that $\mathcal D_\pi$ only contains $s$ at the mode of $P_{\pi^*}(s)$? Wouldn't the RHS be at least as large as the LHS, in that case?

S4.1: $P_E$ is undefined (it's mentioned that it's the expert's state distribution, but a formal definition is needed). Is it more correctly denoted $P_{\pi_E}$, to "typecheck" with the subscripting of $P$ with a policy (i.e. $P_\pi$) to denote the state visitation distribution induced by policy $\pi$?

### Other issues
- The motivation of the gating parameter ($\lambda$) seems to come from dataset size imbalance (between $\mathcal D_E$ and $\mathcal D_{\pi_{\theta_i}}$). Why can't this dataset balance be addressed by simply normalizing the losses by the size of the datasets? Can the authors comment? If this is insufficient, the paper needs to discuss why.
- Missing related work: like the submitted paper, [A] also performs likelihood-based imitation learning with a normalizing flow for expert distribution matching, is not adversarially trained, and uses perturbation of the expert trajectories with gaussian noise. Discussion of the differences is important to contextualize the current paper's contributions, e.g. [A] used the distribution for planning rather than to learn a final policy
- The description of the experiment corresponding to Figs 11, 12, and 13 is lacking. It's not clear what environment(s) the experiment are run on.
- The introduction needs more justification for why DCNF is used in favor of other likelihood-based deep generative modeling approaches. Is continuity necessary? Is denoising necessary? Section 3 talks about the latter (C#2), perhaps incorporate some of this motivation into the introduction.

[A] Rhinehart N, McAllister R, Levine S. Deep imitative models for flexible inference, planning, and control. arXiv preprint arXiv:1810.06544. 2018 Oct 15.


**Summary Of The Paper:**

This paper proposes a straightforward non-adversarially-trained imitation learning method. The goal is to avoid mode collapse and training instability that can occur in some adversarially-trained imitaiton learning methods, while still addressing the distribution shift problem that basic Behavior Cloning is subject to.

The paper:
- Motivates a two-stage learning procedure by starting with a likelihood-based state-action distribution matching objective that would require bi-level optimization
- Presents a lower-bound of the objective in which optimization can occur serially of two components: MLE of the expert's vistation distribution, and then policy optimization over two terms, one of which is the LL of visited states under the the modelled visitation distribution, and the other is the behavior cloning loss.
- Presents a proof that it is a lower bound.
- Proposes a concrete approach to performing this optimization problem by using a denoising conditional normalizing flow for modeling the expert's visitation distribution, where the CNF seems to be motivated because of its ability to model densities of complex distributions, and the denoising is motivated to enable stable training in situations in which the data lie in a lower-dimensional manifold.
- Performs experiments on 10 mujoco environments that illustrate that the proposed approach achieves better mode coverage than some comparable methods, that the denoising is empirically effective for generalization, that the algorithm is more robust than GAIL to variation in learning rate, among other hyperparmaeter variations.

**Summary Of The Review:**

Despite some important issues with method presentation and justification, I think the paper presents a solid method and solid evidence of its utility. The paper can definitely be improved, but as it stands now, I think it's a good contribution.

---

> ### Author Response · Authors · 2021-11-19
> **Response to Reviewer nLuT [Part 1/2]**
>
> We appreciate the reviewer for providing positive comments and constructive suggestions to this work. Given the reviews, we have addressed most questions in the **updated revision**.
>
> #### **W1:** "There are some issues (a,b,c) with the proof. (a) The proof refers to component ${\scriptstyle \enclose{circle}{\kern .06em 2\kern .06em}}$ in Eq 1, yet Eq. 1 has no denoted components. Is this reference correct? If so, then Eq 1. should denote the second component with an underbrace. If not, should it instead be Eq. 2 or Eq. 3?"
>
> **R1:** Thank you for pointing this out. You are correct that the ${\scriptstyle \enclose{circle}{\kern .06em 2\kern .06em}}$ refers to Eq.(2). We have modified it in the updated revision in Appx. A where we have rewritten eq.(2) and eq.(3) together with their denoted parts to make it clear.
>
> #### **W2:** "(b) The proof states "with sufficient expert data $\mathcal{D}\_E$, ${\scriptstyle \enclose{circle}{\kern .06em 2\kern .06em}}$ can be expressed with $\sum_{s\in\mathcal{D}_\pi}\frac{P_E(s)}{P_\pi(s)}\log P_\pi(s)$. I don't see how this is true, because ${\scriptstyle \enclose{circle}{\kern .06em 2\kern .06em}}$ in Eq (3) is the state distribution is the state distribution of the policy. Again, is the reference to ${\scriptstyle \enclose{circle}{\kern .06em 2\kern .06em}}$ correct? These ambiguities / errors prevent me from being able to completely follow the proof."
>
> **R2:** We apologize for the incorrect reference. In the updated revision, the derivation of this equality can be done by applying importance sampling.
>
> #### **W3:** "It's not obvious from the definition of $g(\pi, P_\pi)$ that $\sum_{s\in\mathcal{D}\_{\pi^*}}\log P_{\pi^*}(s)\geq \max_\pi \sum_{s\in\mathcal{D}_\pi}\log P_E(s)$ for two reasons (c.i) because the sums are not normalized by the the dataset size, it would be easy to invalidate the inequality by changing the dataset sizes (c.ii) What if $\pi$ was such that $\mathcal{D}\_\pi$ only contains $s$ at the mode of $P\_{\pi^*}(s)$? Wouldn't the RHS be at least as large as the LHS, in that case?
>
> **R3:** We agree that normalization will make the proof more rigorous, and modified the summation notations to expectation notations in the **updated revision**. This is consistent with our implementation as you can check in the code. Yes, when $\pi$ whose $\mathcal{D}\_\pi$ only contains $s$ at the mode of $P\_{\pi^*}(s)$, the RHS will be at least as large as the LHS which does not conflict with our statement.
>
> #### **W4:**  $P_E$ is undefined.
> **R4:** Thank you for pointing this out. We have added the $P_E$ definition in the 1st Paragraph in Sec. 3 in the updated revision as below:
>
> "We denote $P_\pi(s,a)$ as the probability of observing a state-action pair $(s,a)$ when executing the learner policy $\pi$, and denote $P_E(s,a)$ to represent $P_{\pi_E}(s,a)$ for brevity."
>
> #### **W5:** "The motivation of the gating parameter $\lambda$ seems to come from dataset size imbalance (between
> $\mathcal{D}\_E$ and $\mathcal{D}\_{\pi_{\theta_i}}$). Why can't this dataset balance be addressed by simply normalizing the losses by the size of the datasets? Can the authors comment? If this is insufficient, the paper needs to discuss why."
>
> **R5:** Introducing $\lambda$ to disable RL objective in some training iterations is a practical trick that can be applied to both improve sample efficiency (decrease the number of interactions with the environment) and training stability (as is shown in Fig. 11). Such a trick is can be viewed as a strategy of learning rate control as is used by Lee., et.al., Deeptwist, arXiv:1810.12823, 2019. To make it more clear, we have updated the wording in the updated revision as below:
>
> "In each training iteration, the learning speed between the two components matter, and have impacts on the training stability. Similar observations are made in Lee et al., Deeptwist, 2018."

---

> > ### Author Response · Authors · 2021-11-19
> > **Response to Reviewer nLuT [Part 2/2]**
> >
> > #### **W6:** "Missing related work: Rhinehart N, McAllister R, Levine S. Deep imitative models for flexible inference, planning, and control. [A] also performs likelihood-based imitation learning with a normalizing flow for expert distribution matching, is not adversarially trained, and uses perturbation of the expert trajectories with gaussian noise. Discussion of the differences is important to contextualize the current paper's contributions, e.g. [A] used the distribution for planning rather than to learn a final policy."
> >
> > **R6:** Thank you for pointing out this work. Our proposed SLIL is different from the mentioned work:
> > - **Problem difference:** Our proposed SLIL targets at solving the mode collapse and training instability problems in GAIL. However, Imitative Models (IM) by Rhinehart., et. al., combines IL and planning-based algorithms, and learns a probabilistic model to generate desirable behaviors based on specified goals.
> >
> > - **Technical difference:** SLIL learns expert state distribution using a flow model to learn expert policies. Unlike SLIL, IM trains a flow to learn expert behaviors conditioned on observations, and aims to construct goal distributions with Gaussian prior for better inference of expert behaviors conditioned on goals.
> >
> > We have cited this work in the updated revision.
> >
> > #### **W7:** The description of the experiment corresponding to Figs 11, 12, and 13 is lacking. It's not clear what environment(s) the experiment are run on.
> >
> > **R7:** We have added the task name "the Walker" in the subscripts of these figures for clarification in the updated revision.
> >
> >
> > #### **W8:** "The introduction needs more justification for why DCNF is used in favor of other likelihood-based deep generative modeling approaches. Is continuity necessary? Is denoising necessary? Section 3 talks about the latter (C#2), perhaps incorporate some of this motivation into the introduction."
> >
> > **R8:** Thank you for you constructive suggestion. Indeed, both the continuity and the denoising are necessary. See details below.
> >
> > - **Necessity of CNF:** Expert demonstration data are usually high-dimensional (e.g., with state dimension of 111 for the Ant task and 376 in the Humanoid task), it is necessary to use CNF instead of discrete NF. Continuous NF has the strength over discrete NF as it has flexibility in model architecture design with better generalization ability and scalability to deal with higher dimensional data (Chen et, al., arXiv:1806.07366, 2018, Grathwohl et.al., FFJORD, arXiv:1810.01367, 2018).
> >
> > - **Necessity of denoising:** Expert demonstrations suffers from the manifold hypotheis problem. Adding noise to the CNF model is important to deal with such a problem. In addition, the denoising mechanism helps to smooth the loss landscape of the CNF model to learn more precise expert behavior distributions.
> >
> > We will add the motivation of DCNF in the introduction in the final version of this paper.

---

> > > ### Comment · Reviewer_nLuT · 2021-11-30
> > > **Response to authors**
> > >
> > > Thank you for your response, which has cleared up the concerns I mentioned in my review.

---

> ### Author Response · Authors · 2021-11-28
> **Has our response addressed your concerns?**
>
> Dear reviewer nLuT,
>
> we would be grateful if you can confirm whether our response has addressed your concerns, and let us know if any issues remain. Please take a look at our response, the paper and appendix of the revised version for further details.

---

### Official Review · Reviewer_JhT1 · 2021-11-01

**Correctness:** 3
**Technical Novelty And Significance:** 3
**Empirical Novelty And Significance:** 3
**Recommendation:** 5
**Confidence:** 4

**Main Review:**

$\textbf{Strengths}$
- The approach is very well motivated, aiming to mitigate issues of both BC (distributional shift) and GAIL-like approaches (mode collapse and instability of minimax optimization)
- The algorithm performs very well on all benchmark environments, which are themselves distinct enough.
- The ablations illustrate the robustness of the method to hyperparameter variations.

$\textbf{Weaknesses}$
- In the description of the proposed approach to generative modeling, coined denoising continuous normalizing flows, there are a few points I do not understand, and where their may be some incorrect statements:
  - It is claimed $P_{E,\sigma}(\tilde{s}|s) = P_E(s)\mathcal{N}(\tilde{s}| s, \sigma^2I)$. I think the RHS is actually equal to the joint distribution  $P_{E,\sigma}(\tilde{s}, s)$ instead?
  - The statement that follows about the $0$-noise limit is also incorrect I thinkj, their should be some sort of marginalization in order to have an accurate proof?
  - In the algorithm, I think the noise scale in the Gaussians should be $\sigma_i^2$ instead of $\sigma_j^2$?



**Summary Of The Paper:**

The paper proposes an approach to the problem of state-action based imitation learning, in which an agent aims to solve a specific task given state-action expert trajectories. Their objective SLIL combines (or interpolates between) generative modeling and behavioral cloning in order to avoid mode collapse of the agent policy (common issue of adversarial approaches to IL like GAIL) and to mitigate distributional shift which BC typically suffers from.

Contributions:
- An extension of continuous normalizing flows, coined denoising continuous normalising flows (DCNF), which is claimed to alleviate the manifold hypothesis, and is trained via maximum likelihood estimation.
- A strong imitation learning algorithm consisting in estimating the expert's state distribution, and learning a policy that recovers the expert policy and that has an occupancy distribution close to the distribution induced by generative model learned via the DCNF.


**Summary Of The Review:**

I believe the paper proposes an interesting approach to imitation learning, however the technical issues described in the weaknesses section makes me currently tend towards a weak reject. If the authors were to successfully clarify these, I would consider raising my score.

---

> ### Author Response · Authors · 2021-11-19
> **Response to Reviewer JhT1**
>
> We really appreciate the reviewer for the careful review, based on which we made some clarifications in the updated revision of the paper.
>
> #### **W1:** "It is claimed that $P_{E,\sigma}(\tilde{s}|s)=P_E(s)\mathcal{N}(\tilde{s}|s,\sigma^2 I)$. I think the RHS is actually equal to the joint distribution $P_{E,\sigma}(\tilde{s},s)$ instead? The statement that follows about the 0-noise limit is also incorrect I think, there should be some sort of marginalization in order to have an accurate proof?"
>
> **R1:** We apologize for making it unclear. To make it clear and accurate, we have updated the expression as below at the bottom of page 5:
>
> Denote a perturbed expert state ${s}\sim \mathcal{D}_E$ as $\tilde{s}$, with $\tilde{s}=s+\sigma \epsilon$ where $\epsilon\sim\mathcal{N}(0, I)$. When the noise level is very small, i.e., $\sigma\rightarrow 0$, $\tilde{s}$ approaches $s$.
>
> #### **W2:** "In the algorithm, the noise scale should be $\sigma_i^2$."
>
> **R2:** You are right, we have corrected this typo in the DCNF algorithm. We thank the reviewer again for carefully pointing this out.

---

> ### Author Response · Authors · 2021-11-28
> **Has our response addressed your concerns?**
>
> Dear reviewer JhT1,
>
> we would be grateful if you can confirm whether our response has addressed your concerns, and let us know if any issues remain. Please take a look at our response, the paper and appendix of the revised version for further details.

---

### Official Review · Reviewer_kBN4 · 2021-11-04

**Correctness:** 4
**Technical Novelty And Significance:** 3
**Empirical Novelty And Significance:** 2
**Recommendation:** 5
**Confidence:** 4

**Main Review:**

Regarding the strength of this work, the contribution of this work over given baselines is clear, and I'm satisfied with the detailed explanations and experiments, e.g., comparison with LIL, the justification of using DCNF rather than CNF, how DCNF is trained algorithmically, quantitative tests using EMD and test log likelihoods, etc.

Regarding the weakness of this work, I think some recent relevant works are missed, e.g., Dadashi et al., "Primal Wasserstein Imitation Learning", Kim et al., "Imitation with Neural Density Models", and some references therein. Although these papers didn't aim to solve the mode collapsing directly, their algorithms are much more sample-efficient than GAIL in terms of the number of expert trajectories and the number of environment interactions (i.e., GAIL is no more a SOTA algorithm as stated in this work), and both are relevant to this work in the sense that expert support estimation is used. Primal Wasserstein Imitation Learning (PWIL) shows their algorithm works efficiently even when we have a small number of expert trajectories (1 or 11 trajectories), whereas this work presented empirical results assuming more than 4 expert trajectories (more than 80 expert trajectories for Humanoid). In "Imitation with Neural Density Models", energy-based model is used to estimate expert support estimation, which are based on maximmum likelihood estimation similar to this work.

**Summary Of The Paper:**

This work proposed Stabilized Likelihood-based Imitation Learning (SLIL) which iteratively estimates the expert state distribution by using Denoising Continuous Normalizing Flow (DCNF) and maximizes the policy learning objective in Eq. (3) that matches expert policy and state distribution. Compared to other baselines such as BC, DRIL, GAIL, SLIL was shown to find out better expert state distribution (in terms of seeking multiple modes, Earth Mover's Distance (EMD) while achieving the empirical return close to that of the expert (which is widely used in the imitation learning literature). The strength of using DCNF instead of using Continuous Normalizing Flow (CNF) is well desribed and supported by the experiments evaluating the test log likelihood in Figure 9. Authors also analyze SLIL's hyperparameter sensitivity that supports the claim on the SLIL's stability.


**Summary Of The Review:**

Overall, writing is clear, and the empirical results are shown to outperform GAIL and some other baselines. However, some SOTA algorithms (which I believe is relevant to this work) are missed, and it is unclear that the mode collapsing issue also happens for those relevant works or not. Although I agree that there is an algorithmic novelty for this work, I believe more elaborated results should be given with the aforementioned references.

---

> ### Author Response · Authors · 2021-11-19
> **Response to Reviewer kBN4**
>
> We thank the reviewer for the careful review and pointing out these interesting references. Below we list the differences between these references and our proposed SLIL:
>
> 1. $\textbf{Problem difference:}$ Unlike the PWIL and Imitation with Neural Density Models (NDI) works that target on avoiding adversarial training, our proposed SLIL focuses on dealing with the mode collapse and training stability problems in GAIL.
>
> 2. $\textbf{Mode collapse in PWIL:}$ Regarding PWIL: PWIL applies the primal form of the Wasserstein distance to measure the difference between expert and learner behavior distributions and avoid adversarial training. Though efficient in expert demonstration numbers, this work also suffers from the mode collapse problem as is shown in Fig. 1f (page 2) of the updated revision. In Fig. 1f in the Reacher4 task with four expert modes, PWIL collapse to the blue and yellow modes. We will also include its performance on the HalfCheetah2 task for more experimental support in the final version of this paper.
>
> 3. $\textbf{Regarding NDI:}$ Different from our proposed SLIL which starts from expert state-action distribution maximization, NDI proposes to minimize the (generalized) reverse KL divergence between the occupancy measures of expert's and learner's. Though these two methods all require learning expert behavior distribution as a reward, they stem from different problems and objectives. Considering that the NDI work has just been recently accepted in NeurIPS'21 and its code has not yet been released, we do not add it as a baseline due to the limited time. We are working on implementing it ourselves for comparison.
>
> Based on your comments, we have **updated a revision** where we cited these two works in the related work section.

---

> ### Author Response · Authors · 2021-11-28
> **Has our response addressed your concerns?**
>
> Dear reviewer kBN4,
>
> we would be grateful if you can confirm whether our response has addressed your concerns, and let us know if any issues remain. Please take a look at our response, the paper and appendix of the revised version for further details.

---

### Decision · Program_Chairs · 2022-01-20

**Decision:**

Reject

**Comment:**

The paper proposes the use of a state distribution estimation objective with a classic behavioral cloning objective for imitation learning.
The submission also proposes the use of a continuous normalizing flow training technique coined "denoising normalizing flow" to learn the state distribution. The authors experimentally validate their method on several MuJoCo continuous control benchmarks.
The theorem 4.1 does validate the fact that this proposed objective is can be maximized by the target policy.
However, the technical contributions (proposal of new objective and the denoising normalizing flow method) are marginal compared to previous work (e.g., SoftFlow or Energy-Based Imitation Learning).
The empirical validation is lacking more extensive comparison with PWIL or NDI, which are more recent methods attempting to address the challenges described in the submission.
I'm recommending this paper for rejecting for this conference.